# Differential Neural Network-Based Nonparametric Identification of Eye Response to Enforced Head Motion

**Isaac Chairez [1,2]** , **Arthur Mukhamedov [3]** , **Vladislav Prud [3]** , **Olga Andrianova [4,\*]** and **Viktor Chertopolokhov [5]**

1 Bioprocesses Department, UPIBI, Instituto Politecnico Nacional, Ciudad de Mexico 07340, Mexico; jchairezo@ipn.mx
2 School of Engineering, Tecnologico de Monterrey, Campus Guadalajara, Monterrey 64849, Mexico
3 Center "Supersonic", Lomonosov Moscow State University, 119991 Moscow, Russia; a.mukhamedov@vrmsu.ru (A.M.); info@vrmsu.ru (V.P.)
4 V.A. Trapeznikov Institute of Control Sciences of RAS, 117997 Moscow, Russia
5 Center "Supersonic", Interdisciplinary Scientific and Educational School "Mathematical Methods of Large-Scale Systems Analysis", Lomonosov Moscow State University, 119991 Moscow, Russia; psvr@vrmsu.ru
\* Correspondence: andrianovaog@gmail.com; Tel.: +7-926-888-0832

**Abstract:** Dynamic motion simulators cannot provide the same stimulation of sensory systems as real motion. Hence, only a subset of human senses should be targeted. For simulators providing vestibular stimulus, an automatic bodily function of vestibular–ocular reflex (VOR) can objectively measure how accurate motion simulation is. This requires a model of ocular response to enforced accelerations, an attempt to create which is shown in this paper. The proposed model corresponds to a single-layer spiking differential neural network with its activation functions are based on the dynamic Izhikevich model of neuron dynamics. An experiment is proposed to collect training data corresponding to controlled accelerated motions that produce VOR, measured using an eye-tracking system. The effectiveness of the proposed identification is demonstrated by comparing its performance with a traditional sigmoidal identifier. The proposed model based on dynamic representations of activation functions produces a more accurate approximation of foveal motion as the estimation of mean square error confirms.

**Keywords:** nonparametric model; artificial neural network; Izhikevich artificial neuron; vestibular–ocular reflex; control Lyapunov function

**MSC:** 93B30; 93-10; 93D30; 93C10; 94C30

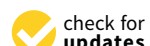



## 1. Introduction

Currently, a significant multidiscipline effort deals with developing technologies that can be applied for training in simulated environments. Such training can be used in different scenarios, from studying drivers' behavior to improving road safety and pilot training, the latter of which has been one of the leading forces for the development of these systems since the early years. These technologies require understanding human sensory systems and their influence to be studied effectively with the proposer instrumentation and modeling tools.

During simulator training, body movements cannot precisely match what is being shown on screen, causing a mismatch in sensory information and leading to simulator sickness as described in [1,2]. This discrepancy is caused by several factors like delays due to tracking and rendering of the output image and physical limitations of the movement range of training systems. Consequently, attempts to overcome this problem covers several different research directions, including but not limited to dynamic motion systems, forecasting movement, and galvanic vestibular stimulation [3]. However, the problem can also be reversed, so that body reaction is used to estimate the accuracy of simulated motion.

One of such indicators is an ocular response to enforced accelerations by an external system or device, just like a flight simulator.

Due to the size and position of the fovea, which is the part of a human eye retina with a high density of light-sensitive photoreceptors, clear vision is achieved when the object of interest is moving slower than $4°/s$. A unique mechanism exists so that the region of interest on the acquired image stays on the retina as the body moves. It is called the vestibular–ocular reflex (VOR), and it is one of the interaction processes between a human body and the surrounding environment. It operates via a neural path between the vestibular and oculomotor systems: eyes compensate head rotations by rotating in the opposite direction [4].

Incorrect functioning of VOR leads to disruptions of clear vision such as the inability to compensate micromovements of the head. However, as an existing connection between external accelerations and angular velocities with the vestibular response is not entirely understood, VOR cannot be estimated directly. A natural way to study VOR is to observe it using immersive technologies (such as virtual or mixed reality) and produce reliable and accurate mathematical models of VOR with human motion as input and electrophysiological response as output. This response could be electroencephalographic signals, oculographic information, or eye motion data, among others. Despite the importance of such mathematical model design, the number and complexity of physiological aspects increase the difficulty of generating specific models for given motion cues that use a reasonably small number of parameters [5].

An alternative way to represent VOR dynamics is to use nonparametric models to reproduce the aforementioned input–output relationship while maintaining a tractable numerical complexity. Several methodologies propose nonparametric models, including adaptive autoregressive systems, polynomial approximations, swarm optimization techniques, and artificial neural networks. Nevertheless, the dynamic nature of VOR limits the applicability of the models under a wide variety of working scenarios. Dynamic approximate models can also be considered as modeling options for systems describing VOR dynamics. In particular, differential neural networks (DNNs) have been used for a long time as efficient modeling strategies of dynamic systems with uncertain mathematical models that are affected by perturbations and modeling inaccuracies. Notice that DNN based models could be well fitted to represent the VOR dynamics [6,7]. Still, the selection of activation functions could be a matter of discussion, considering that sigmoidal or other monotonical functions may not capture the complex electrophysiological VOR response.

Izhikevich model of neuron activity [8] is a bioinspired characterization of electrophysiology-based approximate mathematical models. Izhikevich artificial mathematical models have been proven to be an efficient model of diverse neuron responses [9]. Therefore, an aggregation of several Izhikevich artificial neurons is named electrophysiology-inspired approximated DNN or spiking DNNs [10,11].

Because of the modeling abilities of DNN using Izhikevich neuron dynamics, this paper proposes a method to approximate oculomotor response using the described spiking DNN model. The main contributions of this study can be summarized as follows:

- a novel modeling strategy is proposed for the ocular response on head movements based on a spiking DNN with no parameters;
- a new aggregated system is used to confirm the validity of the proposed model. It consists of an experimental system with a motion platform, inertial sensors, an eye-tracking device for acquiring data, and a neural network for processing it.

This manuscript is organized as follows. In Section 2, we provide a general description of the vestibular–ocular response. In Section 3, we introduce the uncertain model of ocular response, which is then formulated as a spiking-differential-neural-network-based nonparametric identifier in Section 4. In Section 5, we describe general modeling strategy as the process of collecting experimental data. In Section 6, we cover processing of the obtained data and assessing performance of the proposed model. Conclusions and final remarks of Section 7 close the study.

## 2. Description of Vestibular–Ocular Connection

As jet aviation and then crewed spaceflight progressed, they brought attention to several physiological phenomena: a vestibular–ocular reflex. Its disruption was stated to lead to deterioration of a human being in the pioneering work by A.L. Yarbus [12]. Possible causes of disorder include biological prerequisites like vestibular neuronitis [13] or congenital predisposition [14] as well as environmental change. Crewed spaceflight provided an essential context for studying the activity of the vestibular system and its connection to the rest of the body. The papers by I. Kozlovskaya and L. Kornilova (Institute of Biomedical Problems, Moscow, Russia) [15,16] examine vestibular–sensory disorders in a weightless environment and methodology for diagnosing the VOR functioning.

A general approach for detecting dysfunctions is to compare actual data with the reference. For vestibular–sensory disorders, the latter takes the form of a VOR model. The most common method of creating such models is to describe the system as a dynamic one formed by differential and difference equations. One such example is [17] that uses a bilateral model of an eye. It describes ocular dynamics based on the activity of extraocular muscles connected to the right and left sides of an eye. These muscles are more sensitive to positive difference, so they are more active when the difference is negative [18]. The downside of this model is that muscle behavior is described using a large number of parameters that require the application of genetic algorithms to improve the model accuracy [19].

An alternative method was proposed in [20]. It uses statistical methods to approximate the actual dynamics of optokinetic–vestibule–cervical and vestibular nystagmus. Typical dynamics of nystagmus' slow phase drive the values of the five parameters of the model. With known dynamics of head rotations and depending on supporting visual information, this model generates both phases of nystagmus. However, such modeling approaches do not provide enough flexibility and require vast processing power to solve the underlying optimization problem.

## 3. Modeling Ocular Response to Enforced Acceleration

This study is focused on developing a nonparametric model based on a single-layer DNN able to characterize ocular response. The network uses artificial neurons implemented as Izhikevich models, so it operates as a Spiking DNN or SDNN for short. The proposed model produces a vector of two angular coordinates of ocular rotation based on linear acceleration and angular velocities from a vestibular system which serves as an input. Training input data come from a tracking system and ground truth output from a bidimensional eye tracker. The two signals were resampled to have equally acquired information.

Let $\zeta = [x_{eye}, y_{eye}]^\top$ be the coordinates vector of the eye movement. Its evolution over time is forced by information from the vestibular system—linear acceleration $a = [a_x; a_y; a_z]$ and angular velocity $\omega = [\omega_x; \omega_y; \omega_z]^\top$. These values are obtained with respect the body motion.

The electrophysiological system relating inertial information with ocular movement operates using the physiological process of VOR. The continuous dynamics of $\zeta$ as the system state vector, coupled with input vector $u = [a^\top; \omega^\top]^\top$ justifies that a model of this relation has uncertain dynamics defined by the following differential equation:

$$\frac{d}{dt}\zeta(t) = f(\zeta(t), u(t)) + \eta(t). \tag{1}$$

Here $\zeta = \zeta(t)$ is the state vector, $u \in \mathbb{R}^6$ is the input vector that drives uncertain dynamics described by the proposed vector function $f : \mathbb{R}^2 \times \mathbb{R}^6 \to \mathbb{R}^2$. $f$ is Lipschitz with respect to its first argument with a positive constant $L_f > 0$. $\eta \in \mathbb{R}^2$ is the vector of external perturbations to the system not involved in the modeling process. These perturbations belong to a subset of $\Sigma = \{\eta \mid \|\eta\|^2 \leq \eta_0, \eta_0 > 0\}$. Such class is admissible considering the nature of inputs and signals that affect the VOR dynamics.

## 4. Formulation of Spiking-Differential-Neural-Network-Based Model

For the vestibular–ocular system with an uncertain mathematical model (1), the SDNN formulation assumes the following form:

$$\frac{d}{dt}\zeta(t) = A\zeta(t) + W_1^o \phi_1(\zeta(t)) + W_2^o \phi_2(\zeta(t))u(t) + \tilde{f}_e(\zeta(t), t) + \eta(t),$$
$$\zeta(0) = \zeta_0 \in \mathbb{R}^2. \tag{2}$$

The vector $\zeta \in \mathbb{R}^2$ defines the SDNN state. The matrix $A \in \mathbb{R}^{2\times 2}$ describes the linear component of the network dynamics. This matrix is selected as a Hurwitz one to provide boundedness for the state $\zeta$. The two following components form approximation of an uncertain system with traditional SDNN. $W_1^o \in \mathbb{R}^{2\times p_1}$ and $W_2^o \in \mathbb{R}^{2\times p_2}$ are the weights matrices and $\phi_1 : \mathbb{R}^2 \to \mathbb{R}^{p_1}$ and $\phi_2 : \mathbb{R}^2 \to \mathbb{R}^{p_2\times 6}$ are the vector and matrix of activation functions respectively. Choice of the exact values of $p_1$ and $p_2$ is left to the SDNN designer, depending on the value of expected approximation error and methodologies of selecting the size of each layer of general artificial neural networks.

Dynamic nature of the real biological neural networks bioinspired the proposal in this study to use activation functions based on neuron evolution. Thus, each component of $\phi_1$ and $\phi_2$ is described as the output of the Izhikevich model of neuron [8]:

$$\frac{d}{dt}\varrho_i(t) = f_0(\varrho_i(t), \zeta(t)),$$

$$f_0(\varrho_i, \zeta) = \begin{bmatrix} 0.04v_i^2 + 5v_i - u_i + 140 + Z_i^\mathsf{T}\zeta \\ a_i(b_i v_i - u_i) \end{bmatrix}, \varrho_i = \begin{bmatrix} v_i \\ u_i \end{bmatrix}, \tag{3}$$

$$\text{if } v_i \geq 30 \text{ mV, then } \begin{cases} v_i := c_i \\ u_i := u_i + d_i. \end{cases} \tag{4}$$

Here $a_i$, $b_i$ and $c_i$ are the scalar parameters of the Izhikevich model. $\phi_{ji} = [1, 0]\varrho_i$ characterizes the artificial neuron response and is used as the model output in (2). $Z_i \in \mathbb{R}^2$ is a vector of input weights.

Function $\tilde{f}_e(\zeta(t)) : \mathbb{R}^2 \times \mathbb{R} \to \mathbb{R}^2$ in (2) represents approximation error due to selection of a finite number of Izhikevich neurons in the proposed SDNN design. Based on SDNN modeling characteristics this error belongs to the following set: $\Omega = \{\tilde{f}_e \mid \|\tilde{f}_e\|^2 \leq \tilde{f}_0, \tilde{f}_0 > 0\}$. This result is a consequence of the dynamics of the Izhikevich artificial neuron.

The term $\eta \in \mathbb{R}^2$ in (2) characterises external perturbations, or elements affecting VOR system dynamics while being independent of the states values. This term can be said to belong to the set $\Sigma = \{\eta \mid \|\eta\|^2 \leq \eta_0\}$ with $\eta_0$ being a positive scalar. Together, the two terms $\tilde{f}_e$ and $\eta$ represent the degree of vagueness of the underlying electrophysiological system when describing dynamic activation functions of the SDNN representation.

Based on the described approximate dynamical model, this study considers a model for uncertain dynamics of the VOR based on the design of an adaptive SDNN. The proposed approximate adaptive model can be described as follows:

$$\frac{d}{dt}\hat{\zeta}(t) = A\hat{\zeta}(t) + W_1(t)\phi_1(\hat{\zeta}(t)) + W_2(t)\phi_2(\hat{\zeta}(t))u(t), \quad \hat{\zeta}(0) = \hat{\zeta}_0 \in \mathbb{R}^2. \tag{5}$$

Vector $\hat{\zeta}$ defines the approximated dynamics of the 2 eye coordinates. The right-hand side of the VOR dynamics consists of spiking neurons and satisfies the model structure described in (2). The parameters $W_1$ and $W_2$ in (5) must be adjusted by a set of learning laws. It is necessary to have the learning laws derived in such a way so that the proposed identifier operating under these learning laws and identical input can reproduce state trajectories of (1). The aforementioned allows issuing the following problem formulation corresponding to the modeling process based on the application of Izhikevich artificial neurons.

Problem statement for the nonparametric modeling with SDNN.

The problem considered in this study is designing the nonlinear algorithm $\Sigma(x, \hat{x}, x, u)$ adjusting the weights $W = [W_1 \; W_2]$ in a way that ensures the identification error $\Delta = \zeta - \hat{\zeta}$ has a stable equilibrium point at the origin:

$$\limsup_{T \to \infty} \left\{ \sup_{\eta \in \Sigma, \; \tilde{f}_e \in \Omega} \|\Delta(T)\|_P^2 \right\} \leq \gamma \tag{6}$$

where $\gamma > 0$ defines the quality of approximation of the proposed SDNN. $P \in \mathbb{R}^{2 \times 2}$ is a positive definite matrix that adjusts influence of different components of the modeling error vector to the overall approximation quality.

This problem can be solved using Lyapunov stability theory by deriving dynamics of $W_1$ and $W_2$ from identification error $\Delta$. To develop the stability study, the dynamics of $\Delta$ admits the following ordinary differential equation:

$$\begin{aligned} \frac{d}{dt}\Delta(t) &= A\Delta(t) + W_1^* \tilde{\phi}_1\big(\hat{\zeta}(t)\big) + W_2^* \tilde{\phi}_2\big(\hat{\zeta}(t)\big)u(t) + \\ &\quad \tilde{W}_1(t)\phi_1\big(\hat{\zeta}(t)\big) + \tilde{W}_2(t)\phi_2\big(\hat{\zeta}(t)\big)u(t) + \tilde{f}_e(t) + \eta(t). \end{aligned} \tag{7}$$

The process of applying Lyapunov-based stability confirms that identification error has an upper ultimate bound [21,22]. The suggested Lyapunov function has a quadratic form that depends on identification error and SDNN weights. Dynamics of these weights must be selected in such a way to ensure identification error may have an ultimate bound. The following theorem demonstrates that such a bound exists.

**Theorem 1.** *If there exist positive definite matrices $\Lambda_1 > 0$ and $\Lambda_2 > 0$ and positive and bounded scalar $\alpha > 0$ such that for the matrix inequality $Ric(P, \alpha) < 0$*

$$Ric(P, \alpha) := P\left(A + \frac{\alpha}{2}I_{2 \times 2}\right) + \left(A + \frac{\alpha}{2}I_{2 \times 2}\right)^\top P + PRP + Q,$$

$$R := \sum_{j=1}^{2} W_j^+ \left(\Lambda_j^{-1}\right) I_{2 \times 2}, \quad Q := 2I_{2 \times 2} + \sum_{j=1}^{2} L_j \Lambda_j, \tag{8}$$

*there exists at least one positive definite solution $P \in \mathbb{R}^{2 \times 2}$, $P = P^\top > 0$ then the learning laws described by*

$$\frac{d}{dt}W_j(t) = -k_j^{-1}\Omega_j(t) + \alpha \tilde{W}_j(t),$$
$$\Omega_j(t) = P\Delta(t)\phi_j^\top(\hat{\zeta}(t)), \tag{9}$$

$$W_1(0) = W_{1,0}, \quad W_2(0) = W_{2,0}, j = \{1, 2\},$$

*with scalars $k_1, k_2 > 0$, $\tilde{W}_j = W_j^{tr} - W_j$, with $W_j^{tr}$ any matrix satisfying $\|W_j^{tr} - W_j^0\|_F^j \leq W_j^+$ justify the identification error $\Delta$ converging to a ball with its center at the origin and an ultimate bound given by*

$$\gamma \leq \frac{\eta_0 + \tilde{f}_0}{\alpha}. \tag{10}$$

**Proof of Theorem 1.** Taking into consideration the dynamics of the identification error $\Delta$ presented in (7), one may propose an energetic function depending on the deviation between the state $\zeta$ and $\hat{\zeta}$ as well as the deviation between the weights estimated with the identifier and the actual values of the approximation.

For the particular case of the SDNN considered in this study, the aforementioned energetic function is given by:

$$E(\Delta, \tilde{W}_1, \tilde{W}_2) = \|\Delta\|_{2,P}^2 + k_1\|\tilde{W}_1\|_F^2 + k_2\|\tilde{W}_2\|_F^2. \tag{11}$$

Here $\Delta$ is the tracking error already, for which its dynamics has been defined in (7). The symbol $\|\cdot\|_{2,P}^2$ represents the weighted $l_2$ norm of finite-dimensional vectors with the positive definite and symmetric matrix $P \in \mathbb{R}^{2\times 2}$. Additionally, the terms $\|\tilde{W}_j\|_F^2$, $j = 1, 2$ are the matrix norms of the deviation weights $\tilde{W}_j$. For this study, the trace operator is selected as the matrix norms for the weights deviations. Hence, the energetic function is

$$E\left(\Delta, \tilde{W}_1, \tilde{W}_2\right) = \Delta^\top P\Delta + k_1 tr\{\tilde{W}_1^\top \tilde{W}_1\} + k_2 tr\{\tilde{W}_2^\top \tilde{W}_2\}. \tag{12}$$

Notice that the function $E$ operates as a Lyapunov-like class with a positive definite, null value when the three arguments vanish and are radially unbounded. Now, the full-time derivative of $E$ corresponds to

$$\frac{d}{dt}E(t) = 2\Delta^\top(t)P\frac{d}{dt}\Delta(t) + 2k_1 tr\left\{\tilde{W}_1^\top \frac{d}{dt}W_1\right\} + 2k_2 tr\left\{\tilde{W}_2^\top \frac{d}{dt}W_2\right\} \tag{13}$$

where $E(t) := E\left(\Delta(t), \tilde{W}_1(t), \tilde{W}_2(t)\right)$. The term $2\Delta^\top(t)P\frac{d}{dt}\Delta(t)$ admits the following upper bound

$$2\Delta^\top(t)P\frac{d}{dt}\Delta(t) \leq \|\Delta(t)\|_{2,LM(P)}^2 + \gamma + 2k_1 tr\{\tilde{W}_1^\top \Omega_{W,1}(t)\} + 2k_2 tr\{\tilde{W}_2^\top \Omega_{W,2}(t)\} \tag{14}$$

where $LM(P) = PA + A^\top P + PRP + Q$, while the value of $\Omega_{W,1}(t)$ and $\Omega_{W,2}(t)$ have been presented in the learning laws for the proposed identifier.

Transition in (14) was obtained by applying the Young's inequality [21] $YZ^\top + ZY^\top \leq Y\Lambda Y^\top + Z\Lambda^{-1}Z^\top$, which is valid for any $Y \in \mathbb{R}^{r\times s}$, $Z \in \mathbb{R}^{r\times s}$ and any positive definite and symmetric matrix $\Lambda \in \mathbb{R}^{s\times s}$ a number of times. Taking the result in (14) into the right-hand side of the time derivative of $\frac{d}{dt}E(t)$, leads to

$$\frac{d}{dt}E(t) \leq \|\Delta(t)\|_{LM(P)}^2 + \gamma + 2k_1 tr\{\tilde{W}_1^\top \Omega_{W,1}(t)\} + 2k_2 tr\{\tilde{W}_2^\top \Omega_{W,2}(t)\} + \\ 2k_1 tr\left\{\tilde{W}_1^\top \frac{d}{dt}W_1\right\} + 2k_2 tr\left\{\tilde{W}_2^\top \frac{d}{dt}W_2\right\}. \tag{15}$$

With the addition and subtraction of the following terms $\alpha\|\Delta(t)\|_P^2$, $\alpha tr\{\tilde{W}_1^\top \tilde{W}_1\}$ and $\alpha tr\{\tilde{W}_2^\top \tilde{W}_2\}$, the next right hand side holds for the time derivative of $E(t)$

$$\frac{d}{dt}E(t) \leq \|\Delta(t)\|_{Ric(P,\alpha)}^2 + \gamma - \alpha\|\Delta(t)\|_P^2 + \\ 2k_1 tr\{\tilde{W}_1^\top \Omega_{W,1}(t)\} + 2k_2 tr\{\tilde{W}_2^\top \Omega_{W,2}(t)\} + \\ tr\left\{\tilde{W}_1^\top \left(2k_1\frac{d}{dt}W_1 + \alpha k_1\tilde{W}_1\right)\right\} + tr\left\{\tilde{W}_2^\top \left(2k_2\frac{d}{dt}W_2 + \alpha k_2\tilde{W}_2\right)\right\} - \\ \alpha k_1 tr\{\tilde{W}_1^\top \tilde{W}_1\} - \alpha k_2 tr\{\tilde{W}_2^\top \tilde{W}_2\}. \tag{16}$$

Using the learning laws (9) and the matrix inequality (8) presented in the theorem statement, transforms the right-hand side of the derivative of $E$ into

$$\frac{d}{dt}E(t) \leq \gamma - \alpha\|\Delta(t)\|_P^2 - \alpha tr\{k_1\tilde{W}_1^\top \tilde{W}_1\} - \alpha tr\{k_2\tilde{W}_2^\top \tilde{W}_2\}. \tag{17}$$

Using the definition of the Lyapunov yields the following outcome:

$$\frac{d}{dt}E(t) \leq \gamma - \alpha E(t). \tag{18}$$

The integration of these last differential inclusions and following the convergence to an invariant set scheme presented in [21], yields to prove the ultimate boundedness of the identification error as well as the weights. $\square$

The obtained values of $W_1$ and $W_2$ that minimize the expression (6) may be fixed and used further for solving the prediction problem. The scheme of the whole process (identification and prediction) is shown in Figure 1.

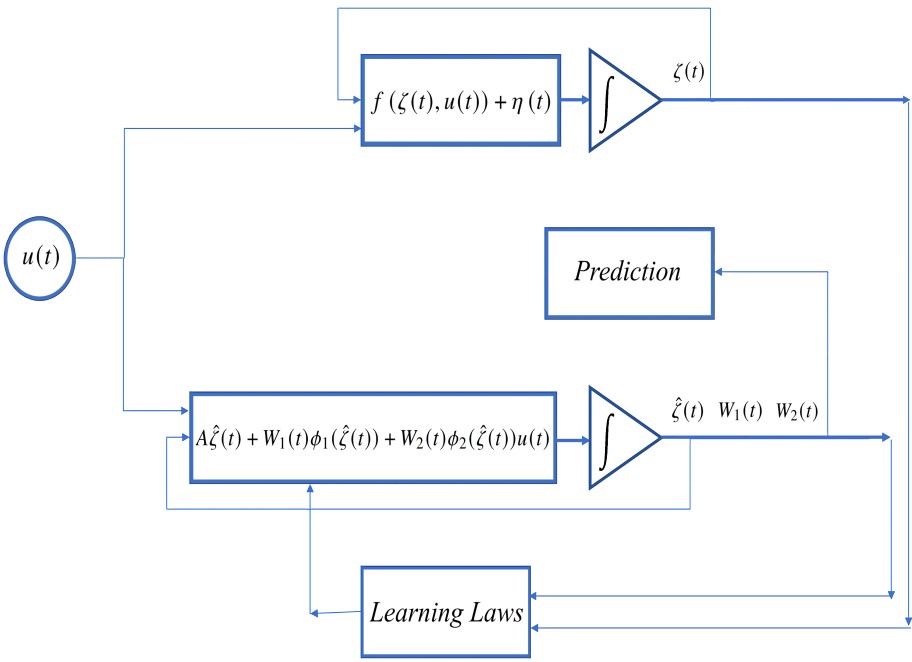

**Figure 1.** Identification and prediction workflow.

## 5. Modeling Process and Experimental Validation

The proposed approximate model was tested in an experiment that collects the data from a volunteer using an instrumented controlled acceleration motion device. The data were recorded at a predefined frequency and then injected (offline) to the proposed SDNN-based identifier. This section details all the aspects of the experiment.

A rotating dynamic platform was used to enforce controlled rotational movements on a test subject. This experiment used an XD-motion platform with 4 degrees of freedom produced by Vympel corporation. The data collecting system is based on a virtual reality headset HTC Vive Pro Eye. The headset's position and orientation quaternion in a fixed coordinate system were obtained from the SteamVR tracking system. SRanipal software gathered data provided by a built-in eye-tracking system and produced view origin and direction vectors for each eye as the output at a maximum frequency of 120 Hz. The whole experimental setup is shown in Figure 2. The resulting ocular movements and head dynamics were recorded and later processed to be modeled by the proposed SDNN.

The experimental process is as follows. First, a test subject puts on and adjusts the belts of the headset for it to stay firmly fixed on the head throughout the whole experiment. Then, the eye tracker is calibrated according to SRanipal documentation and guidelines. After finishing the calibration procedure, any adjustment of the headset by the test subject leads to resetting the experiment, according to SRanipal guidelines. The test subject is then sat on the dynamic platform straight. The platform performs rotational movements around the vertical axis, alternating clockwise and counterclockwise. Movement frequency and amplitude remain constant for 30 s, after which a 20-s break takes place, and new movement parameters are loaded. The order of these parameter sets is randomized. The test subject isn't provided any indication of these parameters. Visual and audio cues of motion are further reduced with the headset screen showing solid black and headphones playing static during the experiment.

The choice of movement pattern is based on several factors. First, horizontal semi-circular channels are stimulated more than the other two for this kind of movement, so ocular response is also primarily horizontal, allowing to focus on a single axis. Second,

the platform has the most reach on this rotational axis, which allows for more diverse movement patterns. Additionally, pitch and roll rotations on this platform are performed by adjusting the length of the legs. However, this adjustment happens even in an idle state when no rotation is being performed, leading to additional platform vibrations introducing parasitic ocular response.

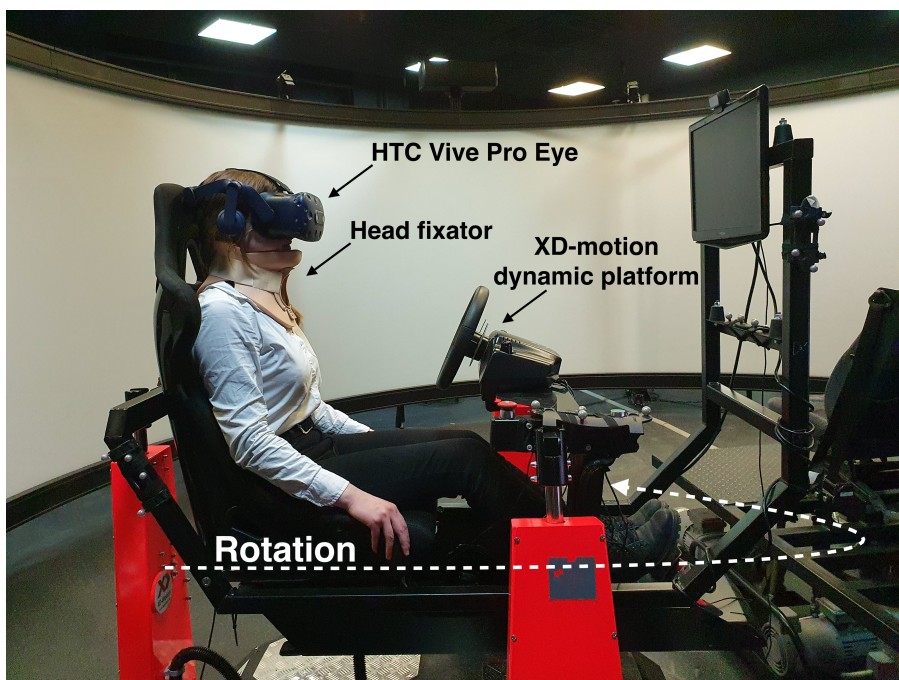

**Figure 2.** Experimental setup for collecting the ocular response to the controlled accelerated movements.

During the processing phase, each movement pattern is handled individually. The leading and trailing 3 s of each recording are trimmed. The view direction vector is converted from a headset coordinate system into angles of eye rotation in horizontal and vertical planes. The head coordinates data were sampled at a lower frequency than eye-tracking data, so the former were smoothed using a Gaussian filter. Head orientation quaternion was converted into Euler angles. After leaving only data corresponding to horizontal angles, angular velocity and linear acceleration were calculated.

### 6. Numerical Simulation

The collected data from the two motion patterns were used to test the proposed SDNN model. These two patterns are 18 25-degree rotation cycles per minute and 50-degree rotations at a rate of 4.8 cycles per minute. They are later referred to as high- and low-frequency movements. As described earlier, linear accelerations and angular velocities formed the system input $u$ while eye rotation angles were used as a reference state $\zeta$. Figures 3–6 compare dynamics of the proposed SDNN identifier with Izhikevich and sigmoidal activation functions on the obtained data. Figures 3a and 5a demonstrate recorded head rotation profile. Figures 3b and 5b show evolution of identification error (shown as mean square error) of the proposed identifier. In both cases, the origin is shown to be a practical stable equilibrium point for the analyzed modeling error. Direct comparison between recorded and modeled data is shown in Figures 3c and 5c. Finally, Figures 3d and 5d show evolution of the weights from initial conditions. The highlighted dashed line on both figures illustrates the work of VOR. The correspondence between ground truth eye-tracking data and identifier state shows the validity of the proposed identifier.

The identification performance of the proposed spiking identifier was compared against the traditional sigmoidal DNN-based identifier, shown in Figures 4 and 6. These figures are structured identically to Figures 3 and 5. Note the different $y$-axis scales

between all figures on the weights dynamics plot. Parameter values for both identifiers are presented in Table 1. Numerical values are compared in Table 2 as the performance of the two approaches using mean square error (MSE), mean absolute error (MAE), and standardized mean absolute error (sMAE).

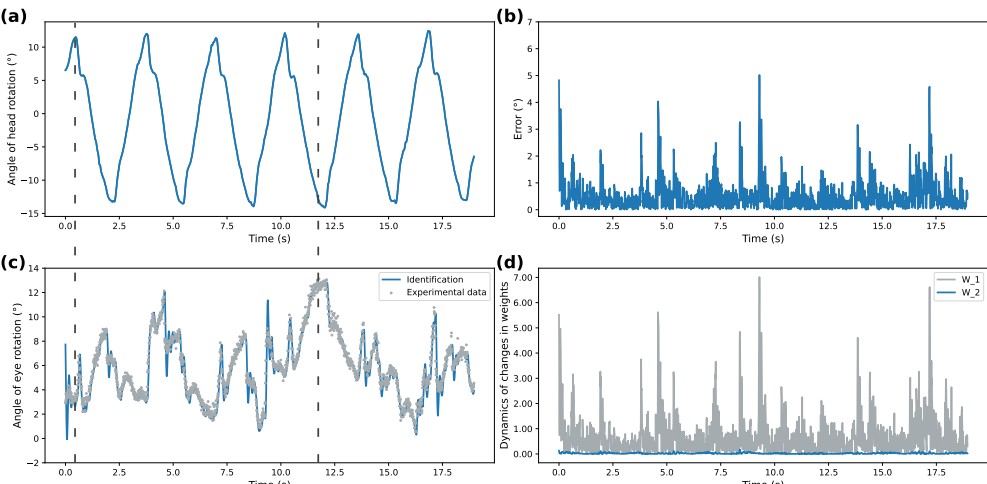

**Figure 3.** Identification with Izhikevich activation function for high-frequency rotations: (**a**)—recorded head rotation; (**b**)—identification error; (**c**)—recorded data and identification results comparison; (**d**)—evolution of weights.

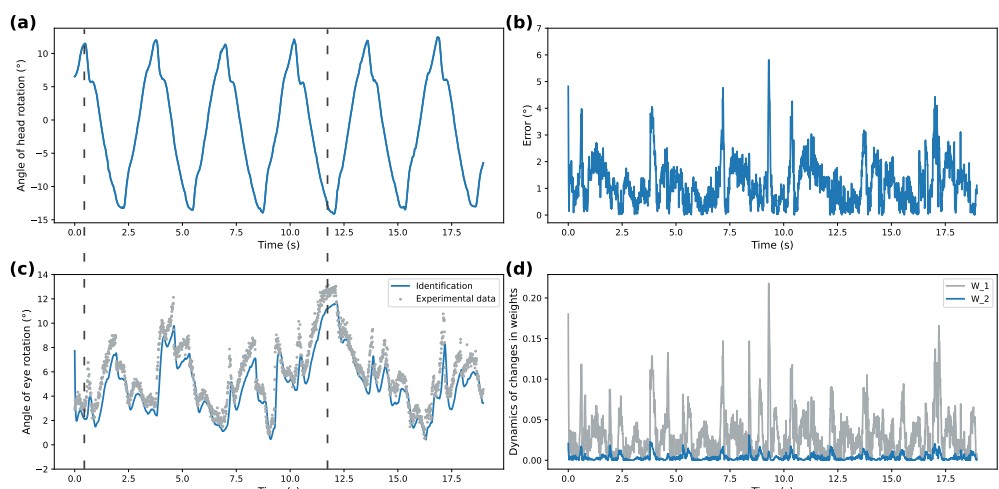

**Figure 4.** Identification with sigmoidal activation function for high-frequency rotations: (**a**)—recorded head rotation; (**b**)—identification error; (**c**)—recorded data and identification results comparison; (**d**)—evolution of weights.

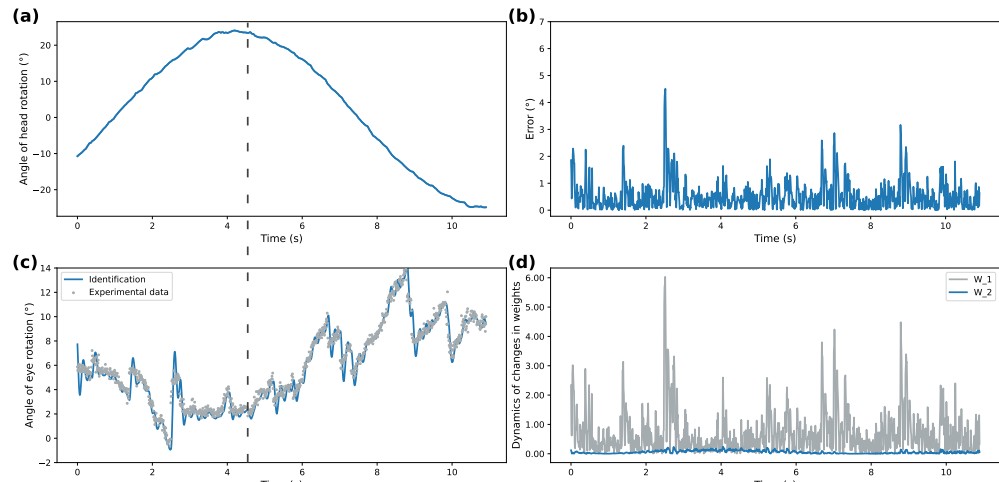

**Figure 5.** Identification with Izhikevich activation function for low-frequency rotations: (**a**)—recorded head rotation; (**b**)—identification error; (**c**)—recorded data and identification results comparison; (**d**)—evolution of weights.

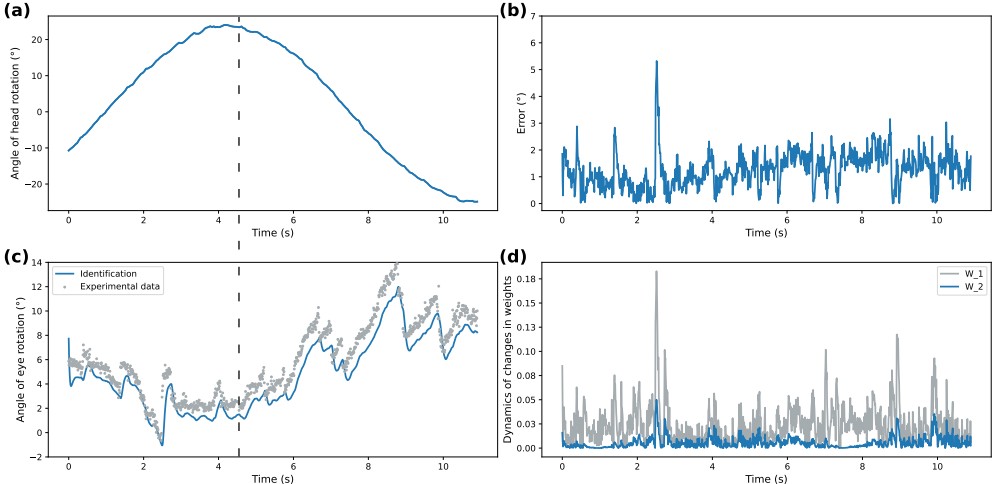

**Figure 6.** Identification with sigmoidal activation function for low-frequency rotations: (**a**)—recorded head rotation; (**b**)—identification error; (**c**)—recorded data and identification results comparison; (**d**)—evolution of weights.

**Table 1.** Parameters of the compared identifiers.

| Parameter | Izhikevich | Sigmoidal |
|---|---|---|
| Matrix $A$ | $20 \times diag(-1, -2)$ | $20 \times diag(-2, -2)$ |
| Matrix $P$ | $1575.9 \times diag(60, 40)$ | $1575.9 \times diag(60, 40)$ |
| Matrix $K_1$ | $0.15 \times diag(10, 1)$ | $0.0001 \times diag(20, 10)$ |
| Matrix $K_2$ | $0.15 \times diag(1, 1)$ | $0.0001 \times diag(20, 10)$ |
| Matrix $W_1(0)$ | $20 \times \begin{bmatrix} 1 & 1 \\ 1 & 1 \end{bmatrix}$ | $0.1 \times \begin{bmatrix} 1 & 1 \\ 1 & 1 \end{bmatrix}$ |
| Matrix $W_2(0)$ | $20 \times \begin{bmatrix} 1 & 1 \\ 1 & 1 \end{bmatrix}$ | $20 \times \begin{bmatrix} 1 & 1 \\ 1 & 1 \end{bmatrix}$ |

**Table 2.** Comparison of identification performance.

| Identifier Type | High-Frequency Data | | | Low-Frequency Data | | |
|---|---|---|---|---|---|---|
| | MSE | MAE | sMAE | MSE | MAE | sMAE |
| Izhikevich | 0.000186 | 0.008948 | 0.119975 | 0.000187 | 0.009647 | 0.140333 |
| Sigmoidal | 0.000710 | 0.021099 | 0.282897 | 0.000588 | 0.021496 | 0.312143 |

Overall, correspondence between modeled behavior and ground truth data shows the applicability of the proposed system under different patterns of rotational movements. Additionally, Izhikevich activation functions for both patterns demonstrate over 50% better performance for modeling ocular response than the DNN implementing sigmoidal activation functions. This shows that SDNN can be used as a generalized approximation class for ocular response dynamics.

## 7. Conclusions

This study examines modeling physiological VOR systems using SDNN. The proposed nonparametric model implements an arrangement of the artificial neurons described by Izhikevich dynamics with fixed parameters to follow eye movements caused by known head accelerations. Learning laws have been derived for the proposed SDNN to ensure convergence to the origin of identification error. An experimental setup is proposed and used to obtain data and confirm the validity of the proposed SDNN-based nonparametric model. Comparison of the proposed modeling strategy and a traditional identifier with sigmoidal activation functions was performed for different experimental conditions and demonstrated the efficacy of the proposed approach. One potential use of this study is estimating the accuracy of motion cues simulation. Suppose the ground truth of the ocular motion is acquired using a model of vestibular–ocular response. In that case, it can be compared with experimental data on a dynamic platform to assess how accurate the movement was in terms of vestibule system reaction. Despite the additional computational complexity produced with the application of Izhikevich models, the identification quality improves significantly compared to the traditional sigmoidal (algebraic form) forms. This fact justifies the approximated model proposed in this study and opens novel options to create representations of complex biological systems with multirate dynamics.

## 8. Patents

A derivative from this work is currently undergoing software registration process.

**Author Contributions:** Conceptualization, I.C., O.A. and V.C.; methodology, I.C. and O.A.; software, V.P. and A.M.; validation, A.M. and V.P.; formal analysis, O.A. and I.C.; investigation, I.C.; resources, V.C.; data curation, A.M.; writing—original draft preparation, A.M. and V.P.; writing—review and editing, I.C., O.A. and V.C.; visualization, V.P.; supervision, I.C.; project administration, V.C. All authors have read and agreed to the published version of the manuscript.

**Funding:** This research was funded by Ministry of Science and Higher Education of the Russian Federation grant number 075-15-2020-923 "Supersonic".

**Institutional Review Board Statement:** Ethical review and approval were waived for this study, due to the study only considered to evaluation of motion cues with volunteers in their normal conditions.

**Informed Consent Statement:** Informed consent was obtained from all subjects involved in the study.

**Data Availability Statement:** Publicly available datasets were analyzed in this study. This data can be found here: https://github.com/cut4cut/spikennet/tree/main/data accessed on 1 February 2022.

**Acknowledgments:** The authors thank Alexander Poznyak and Vladimir Alexandrov for fruitful discussions and helpful suggestions and Ernest Sleptsov for valuable advices concerning literature review.

**Conflicts of Interest:** The authors declare no conflict of interest. The funders had no role in the design of the study; in the collection, analyses, or interpretation of data; in the writing of the manuscript, or in the decision to publish the results.

## Abbreviations

The following abbreviations are used in this manuscript:

| | |
|---|---|
| VOR | Vestibular–Ocular Reflex |
| DNN | Differential Neural Network |
| SDNN | Spiking Differential Neural Network |
| MSE | Mean Square Error |
| MAE | Mean Absolute Error |
| sMAE | Standardized Mean Absolute Error |

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
