# Peer review of "Differential Neural Network-Based Nonparametric Identification of Eye Response to Enforced Head Motion"

_mathematics, doi:10.3390/math10060855_

Round 1
Reviewer 1 Report
Both readability and presentation of the paper should be checked and improved.
It is suggested to discuss more about the findings of this study in the abstract. What are the limitations behind this study? This topic should be highlighted somewhere in the text of manuscript.
The original contribution of the paper is not clearly stated. The contributions of the article must be emphasized in terms of originality, significance, and performance metrics in the abstract and introduction.
Recent studies from high impact factor journal (see https://www.scimagojr.com/) should be cited like from IEEE transactions, Springer and Elsevier in the introduction or in a related work section.
A systematic review of the recent literature could be presented in the related work section.
Acronyms and variables in equations must be defined in the article.
All the mathematical notations should be italicized.
Some figures are with low resolution. Please verify it.
Conclusion: What are the advantages and disadvantages of this study compared to the existing studies in this area?
Reviewer 2 Report
This study is a very current topic. This law was combined with artificial neural networks and a non-parametric analysis was made. I kinly suggest correcting the following for work.
1- Both the abstract and the conclusion part of the study are not sufficient. These sections should be developed and the purpose and results of the study should be clearly emphasized.
2- All details are given for ANN model (table success rate etc.).
3- The work done in the study should be shown as a flowchart and each step should be explained in detail.
4- The references section of the study is weak, and there are too many references in this field.
Round 2
Reviewer 2 Report
Acceptance of the study is appropriate by me